# Development of a novel eye drop aid and evaluation of its efficacy

Yuka Kasai[1], Masako Sakamoto[1], Yuji Matsuda[2], Yuka Ito[3], Hirotaka Haro[2], Kenji Kashiwagi [1]*

1 Department of Ophthalmology, Faculty of Medicine, University of Yamanashi, Chuo, Yamanashi, Japan, 2 Division of Rehabilitation, University of Yamanashi Hospital, Chuo, Japan, 3 Department of Nursing, University of Yamanashi Hospital, Chuo, Japan

* kenjik@yamanashi.ac.jp

## Abstract

### Purpose

The aim of this study was to develop a new eye drop aid and investigate its efficacy in glaucoma patients.

### Methods

The developed eye drop aid is compatible with most therapeutic eye drops and helps users with limited neck retroversion. Adult glaucoma patients admitted to the University of Yamanashi Hospital for glaucoma surgery who fulfilled the following criteria were included: a history of using glaucoma eye drops for at least 6 months, no upper limb loss or dysfunction, and no physical or cognitive disturbances that would interfere with activities of daily living. In terms of corrected visual acuity, the included patients had a logMAR score of at least 1 in at least one eye. Eye drops containing a 0.1% sodium hyaluronate ophthalmic mixture were used in this study. The eye drops were applied to the eyes in the seated position as they would normally be used. The use of the eye drop aid was explained, and the patients tried to instill eye drops with the eye drop aid after two trials. The patients then instilled one eye drop in the supine position, first without the eye drop aid and then with the eye drop aid. Success was defined as the eye drops being dropped into the conjunctival cul-de-sac without touching the ocular surface, face or eyelashes. The patients were surveyed about the usefulness of the eye drop aid. Although the aid was designed to require minimal grip force, we measured the force needed to dispense a single drop for five commonly used clinical eye drops.

### Results

Eighty-eight patients were included (51 men and 37 women, mean age: 67.3±13.4 years). The success rate in the seated position without the eye drop aid was 71.6%,

**Data availability statement:** ll relevant data are within the manuscript.

**Funding:** The author(s) received no specific funding for this work.

**Competing interests:** Kenji Kashiwagi holds a domestic patent in Japan related to the subject of this study. All other authors declare no conflicts of interest. This does not alter our adherence to PLOS ONE policies on sharing data and materials.

and this rate decreased with increasing age; with the eye drop aid, the success rate improved significantly to 97.8%. The success rate in the supine position without the aid (86.4%) was significantly better than that in the seated position, and the eye drop aid increased the success rate to 97.8%. The eye drop aid reduced the squeeze force required to instill eye drops (reduction of 10.3%−53.5%) for all types of eye drops.

## Conclusion

Even though the patients were accustomed to using glaucoma eye drops, 28.4% of the patients were not able to instill eye drops properly. The eye drop aid significantly improved the success rate of eye drop instillation, especially when the eye drops were administered while the patients were seated.

## Introduction

Glaucoma is one of the most common diseases that causes severe visual impairment. Aging is an important risk factor for glaucoma, and the number of glaucoma patients is expected to increase as society ages [1]. Lowering intraocular pressure (IOP) is considered the most effective way to control the progression of glaucoma, and drug treatment with IOP-lowering agents is the mainstay of glaucoma treatment. Patients with glaucoma tend to require an increasing number of ocular hypotensive medications as the duration of their treatment increases [1,2]. Recent studies have highlighted several issues associated with glaucoma treatment, including poor adherence, limited treatment persistence, and medication-related side effects [3–7].

Anti-glaucoma eye drops may be less effective if they are not accurately instilled into the conjunctival sac. Inappropriate usage, including periocular application, excessive dosing, or contamination of the container tip, is associated with an increased risk of adverse events. The proper use of eye drops is one of the most important factors for the effective and safe treatment of glaucoma.

Many patients fail to instill eye drops correctly [8–10]. A variety of factors have been reported as causes of eye drop failure. In a meta-analysis of eye drop failure performed by Davis et al., 18.2–80% of patients contaminated the drop container via contact with the eye or face, 11.3–60.6% released too many drops, and 6.8–37.3% applied the drops to other parts of the eye at the time of ophthalmic administration [11]. In our previous study, the failure rate of ophthalmic drops in glaucoma patients was as high as 61.2% [9]. The important risk factors for eye drop failure are aging, a reduced cervical extension angle, a reduced ability to pick up the eye drop container, reduced motor function of the upper limbs, motor paralysis and a reduced visual field or vision. Consequently, many eye drop instillation aids have been developed [12–16].

Dadak et al. summarized several techniques and aids that improved the eye drop instillation success rate [17]. Davies et al. noted that eye drop instillation aids should be low cost, easy to use, reusable, and compatible with most eye drop bottles [18].

Given this context, we developed an eye drop aid to support successful eye drop placement and investigated the usefulness of and challenges associated with this newly developed ophthalmic aid in glaucoma patients in the present study.

## Patients and methods

The study was performed in accordance with the Declaration of Helsinki. All the participants provided written informed consent. The Ethics Committee of the University of Yamanashi School of Medicine (Approval Code 1574) approved this study. The volunteers who appear in Fig 2 agreed for their images to be included in the paper.

### The developed eye drop aid (S1 Video)

Fig 1 shows an overhead view of the eye drop aid developed and used in this study. Fig 2 and a demonstration video show how to attach the ophthalmic bottles to the aid and then use it for instillation of eye drops. The lid of the eye drop bottle is removed, and the aid is attached. The eye drop aid is held so that it is in contact with the face at two points: the eyebrow area and the upper cheek area. The length of the buccal support that comes into contact with the upper cheek area can be adjusted so that the eye drop bottle is vertical. The eye drop aid has a wing-shaped grasping section that allows a single drop of ophthalmic solution to be released with a single, normal motion. The entire eye drop aid is made of heat-resistant material that can be disassembled and sterilized by boiling. The aid can assist with impaired neck retroflexion because the angle of contact with the face is not vertical. Except for eye drops provided in bottles containing a filter mechanism, which are incompatible owing to the device's adjustable ring, this eye drop aid is compatible with the majority of commercially available glaucoma medications.

### Patients

This study was performed from 1st March 2018–30th October 2019. Adult glaucoma patients who were admitted to the University of Yamanashi Hospital for glaucoma surgery and had a history of at least 6 months of self-administered glaucoma eye drop use, no upper limb loss or dysfunction, and no physical or cognitive disturbances that would interfere with

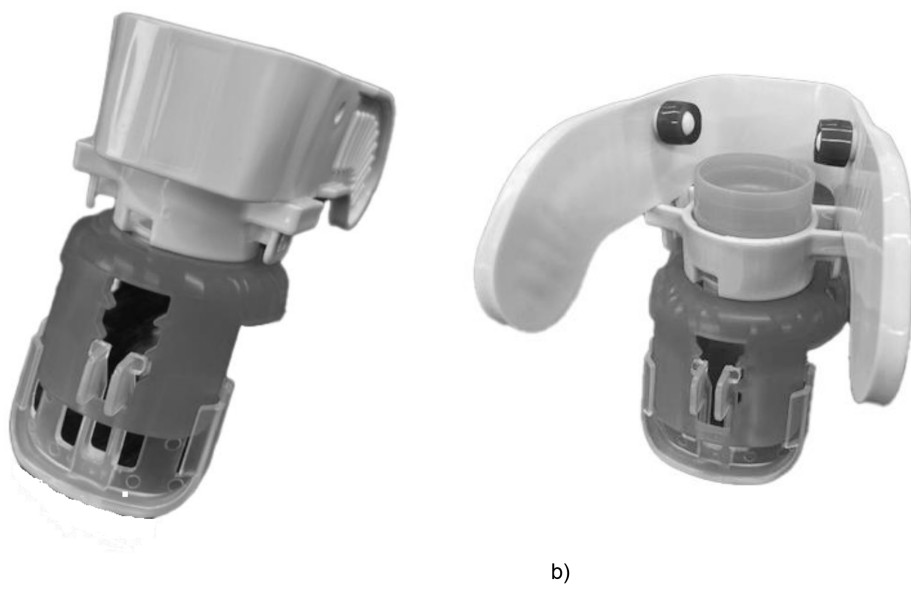

a) b)

**Fig 1. Overhead view of the eye drop aid.** (a) Front side, (b) back side.

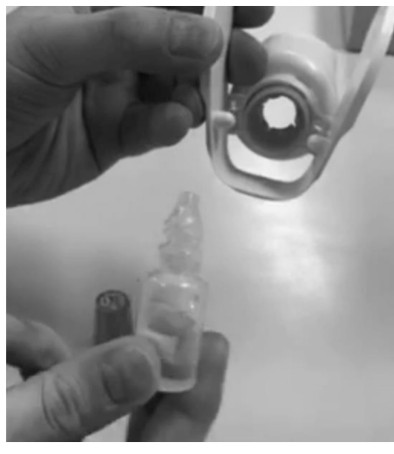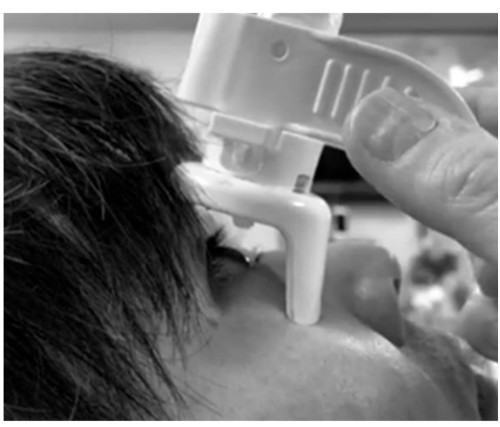

a)                                                    b)

**Fig 2. Procedure for attaching the eye drop bottle and instilling eye drops.** (a) Attaching the eye drop bottle, (b) applying the eye drops. *Note: The image shows the prototype of the developed product. In the final version, the length of the buccal support is adjustable.*

daily life were included. With respect to corrected visual acuity, the patients had a logMAR score of at least 1 in at least one eye. All patients underwent a comprehensive ophthalmological examination, which included an evaluation of best corrected visual acuity, a slit-lamp examination and a fundus examination, within one month prior to admission. Visual field testing was performed within three months prior to the start of the study via the Humphrey Visual Field test program 24−2 (Carl Zeiss Meditec, Inc., CA).

## Experimental design and definition of instillation failure

A 0.1% hyaluronic acid ophthalmic solution (Santen Pharmaceutical Co., Ltd., Osaka, Japan) was used to assess the success of eye drop application. Patients were instructed to administer only one drop, following their usual method. The patient first applied an eye drop in their usual way while in a seated. After the use of the eye drop aid was explained, the patients attempted to use the aid to instill eye drops two times while they were in a seated position. Next, the patients attempted to administer eye drops in the supine position without the eye drop aid, followed by administration with the eye drop aid. Because the majority of patients were accustomed to instilling eye drops in the sitting position, two trials were conducted, one with the eye drop aid and one without it.

A single evaluator (Y.I.) assessed the success of eye drop instillation. Instillation failure was defined as one or more of the following conditions: the tip of the eye drop bottle directly touched the surface of the eye, eyelid, eyelashes or face; or more than two eye drops were applied in a single trial.

## Changes in squeezing force with the eye drop aid

We investigated the extent to which the squeezing force required to apply a single drop of 0.005% Xalatan®(Viatris Pharmaceutical Co., Ltd., Tokyo, Japan), 0.1% AIPHAGAN® (Senju Pharmaceutical Co., Ltd., Osaka, Japan), 0.4% Glanatec® (Kowa Pharmaceutical Co. Ltd, Tokyo, Japan), or 0.1% hyaluronic acid ophthalmic solution (Santen Pharmaceutical Co, Ltd), which are often used as eye drops for glaucoma treatment, changed when the eye drop aid was used compared with application without the aid. We employed a previously reported method to investigate the squeezing force required to apply one eye drop [19]. In brief, either the eye drop bottle alone or the bottle fitted with the eye drop aid was placed into the measurement system, and the squeezing force required to release the first drop was recorded. The squeezing force

measurement was repeated five times, and the average of three values, excluding the highest and lowest measurements, was used as the final squeezing force.

## Statistical analysis

We compared the investigated factors between the successful instillation group and the instillation failure group via the Mann–Whitney U test for continuous variables and Fisher's test for categorical variables. P values less than 0.05 were considered to indicate significance. The corrected visual acuity was converted to a logMAR score. The results are presented as the means ± standard deviations.

## Results

Eighty-eight patients, 51 men and 37 women, with a mean age of 67.3 ± 13.4 years, were included in the study (Table 1).

### Success rate of eye drop instillation without the eye drop aid in the seated position

In the seated position, the overall success rate of eye drop instillation was 71.6% (63/88 patients). The mean age of the successful instillation group was 64.5 ± 14.0 years, whereas the instillation failure group had a mean age of 74.4 ± 7.94 years; this difference was significant (p = 0.01, Mann–Whitney U test). The successful instillation rate was 70.6% (51/87) in men and 73.0% (27/37) in women, with no significant difference (p = 0.806, Fisher's exact test). Age-group analysis revealed a 100% success rate for eye drop instillation among subjects in their 20s and 40s, whereas the success rate declined to 56.7% among subjects in their 70s and 57.1% among subjects in their 80s, indicating a decrease in the successful instillation rate with age (Fig 3).

### Factors contributing to instillation failure without the eye drop aid

Among the 25 patients in the instillation failure group, 12 (48%) experienced eye drops landing around the eye, whereas in 6 patients (24%), the tip of the eye drop bottle made contact with the eyelashes. Seven patients met two or more of the failure criteria. No patients administered more than two drops at a time.

**Table 1. Demographics of the enrolled patients.**

| Male (%) | 51 (58.0) |
|---|---|
| Age (mean ± SD) (yrs) | 67.3 ± 13.4 |
| logMAR (mean ± SD) | Right: 0.467 ± 0.604; Left: 0.6123 ± 0.525 |
| HFA MD (dB) | Right: −14.3 ± 8.6; Left: −15.8 ± 9.6 |
| Age group (yrs) | Number of patients (%) |
| 20–29 | 3 (3.4) |
| 30–39 | 0 (0.0) |
| 40–49 | 5 (5.7) |
| 50–59 | 14 (16.0) |
| 60–69 | 22 (25.0)) |
| 70–79 | 30 (34.1) |
| 80+ | 14 (15.9) |

logMAR: minimum angle of resolution, HFA: Humphrey field analyzer, SD: standard deviation.

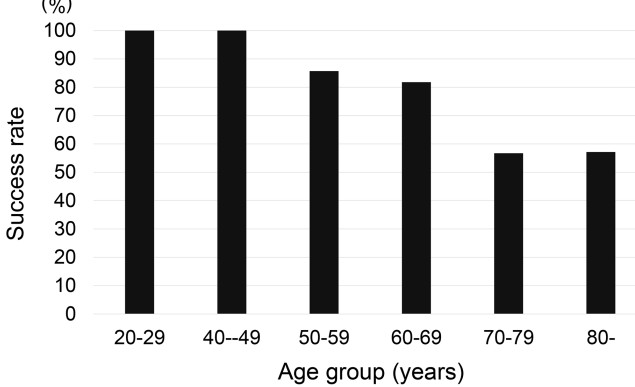

**Fig 3. Comparison of eye drop installation success rates across different age groups.**

### Effect of the eye drop aid on the eye drop installation success rate in the seated position

The eye drop aid significantly improved the overall success rate of eye drop installation to 94.3% (83/88) (P < 0.001, Fisher's exact test). Since the success rate of eye drop installation decreased with age, we evaluated the effectiveness of the eye drop aid in two age groups: individuals under 60 years and those aged 60 years and older. Among those under 60 years of age, the success rate of eye drop installation improved from 83.3% to 97.2% with the use of the eye drop aid, representing an increase of 13.9%. In contrast, in the group aged 60 years and older, the success rate increased from 56.8% to 97.7%, corresponding to an improvement of 40.9%. The difference in improvement rates between the two age groups was statistically significant (p < 0.001, Fisher's exact test). Five patients experienced installation failure due to the administration of eye drops to the periocular area. None of the patients instilled more than two drops in a single attempt.

### Comparison of the success rates of eye drop installation in the supine position with and without the eye drop aid

The success rates of eye drop installation in the supine position without the eye drop aid were 86.4% (76/88) in the first trial and 94.2% (83/88) in the second trial, both of which were significantly higher than those in the sitting position (P < 0.001, Fisher's exact test). Regardless of whether the eye drop aid was used, the success rate in the second trial was significantly higher than that in the first trial. With the eye drop aid, the success rates for the first and second tests were 86.4% (76/88) and 97.7% (86/88), respectively, with a significant improvement in the second trial. The success rate of the second trial with the eye drop aid was greater than that without the aid, but the difference was not statistically significant (p = 0.28, Fisher's exact test).

### Changes in squeezing force with the eye drop aid

The eye drop aid reduced the squeezing force required for installation across all types of eye drops. The greatest reduction was observed with 0.1% AIPHAGAN®, which decreased from 11.1 N to 6.0 N, a 45.9% reduction, followed by 0.1% fluorometholone (7.52 N to 5.0 N, 33.5% reduction), 0.005% Xalatan® (5.1 N to 3.9 N, 23.5% reduction), and 0.4% Glanatec® (6.9 N to 6.1 N, 11.6% reduction).

### Discussion

Treatment using eye drops is a common choice for many ocular diseases. In particular, eye drops are the mainstay of glaucoma treatment; however, previous studies have reported many cases of eye drop failure [8,9]. An et al. reported that many patients fail to use postoperative medication after cataract surgery [20], which could lead to problems such as

postoperative infection and prolonged inflammation. In addition, the use of multiple drops per eye drop instillation and the application of eye drops around the eye may also increase the incidence of side effects.

The eye drop aid developed in this study significantly improved the success rate of eye drop instillation among glaucoma patients. This improvement was greater in elderly patients, who have a higher eye drop instillation failure rate.

Some reasons for eye drop instillation failure, including an insufficient cervical spine extension angle and visual dysfunction, have been identified [8]. In addition, we identified a shallow cervical spine extension angle, weak pinching strength, poor motor dysfunction of the upper limbs, and a degree of ataxia as possible causes of eye drop instillation failure [9].

The possible reasons for the increased success rates of eye drop instillation observed with this aid include the following. The ability to change the length of the brow and cheek supports allows vertical placement of the eye drop bottle even in patients with shallow cervical spine extension angles. The eye drop aid is equipped with a winged design to facilitate eye drop instillation with reduced force. In this study, the squeezing force required to dispense a single drop was reduced by up to 50% across all tested eye drops. We previously reported that the squeezing force required to apply a single drop varies greatly across eye drop bottles [19]. In the present study, the squeezing force required to apply an eye drop varied considerably among eye drop bottles, but the force required to apply one drop was approximately 1–6 N when the eye drop aid was used. This finding indicates that when using this aid, the squeezing force required to apply a single drop does not vary significantly among bottles, and the drops may be easier to apply. Furthermore, no patient in the present study instilled more than one drop per trial when using the aid. This may be beneficial in reducing side effects.

In this study, the improvement in the supine position was smaller than the improvement in the seated position. The reason for this may be that the success rate of eye drop instillation without the aid is better in the supine position than in the seated position. It has been previously reported that posture is related to the success of eye drop instillation, and Naito et al. reported that the supine position is best for successful eye drop placement [21]. While the supine position is generally associated with a higher success rate of eye drop instillation, the participants in this study habitually applied eye drops in the sitting position. Accordingly, this eye drop aid may offer greater benefit for individuals who routinely instill eye drops while seated.

There have been several reports on eye drop aids in the past. Xal-Ease, a dedicated eye drop aid designed for latanoprost eye drops, has been reported to be useful [16]. Gomes et al. reported that this aid prevented contact between the eye drop bottle and the eye or periocular tissue but did not improve success rates for eye drop instillation [22]. In addition, this aid cannot be used with other types of eye drops.

Brand et al. evaluated two types of ophthalmic aids, Opticare and Autodrop, and reported that both were effective in preventing eye drop contamination and ensuring an appropriate drop volume [23].

Davies et al. reviewed eye drop aids and noted that it is desirable for eye drop instillation aids to be low cost, easy to use, reusable, and compatible with most eye drop bottles [18]. Unfortunately, no aids currently exist that fully satisfy these requirements for eye drop aids. The eye drop aid developed in this study is now commercially available in Japan for approximately US$10. This aid is reusable, easily disassembled, and can withstand sterilization by boiling, thus minimizing hygiene issues. It can be used with many commercially available eye drops, including glaucoma eye drops. In addition, the present study confirmed that the success rate of eye drop instillation can be improved with the use of this aid, as the tip of the eye drop bottle does not touch the ocular surface or the surrounding area and it ensures that only one drop is applied at a time. This suggests that the developed eye drop aid meets the requirements identified by Davies et al.

The currently developed eye drop aid has several limitations. In this study, the instillation success rate was higher in the second trial than in the first trial, despite prior explanations of how to use the aid. This demonstrates the importance of teaching correct usage and obtaining a sufficient understanding of this eye drop aid. Although this eye drop aid can be used for many types of eye drops, it currently cannot be used for eye drops provided in a bottle with a filter for preventing contamination or for single-use eye drops. We are considering the development of an eye drop aid suitable for these types of eye drops.

Although we investigated the efficacy of the eye drop aid using only one type of eye drop in this study, many patients are prescribed multiple types of eye drops for daily use [1]. In such cases, patients will need to repeatedly remove and reattach the different eye drop bottles to the eye drop aid each time they instill their prescribed eye drops. Future studies are needed to determine how much of a clinical challenge this will be.

Although most patients in this study were able to correctly administer eye drops with the use of the eye drop aid, some patients were unable to do so. For such patients, self-instillation using only an eye drop aid may be challenging, and alternative treatment approaches may need to be considered. In this study, we selected patients who had been self-administering eye drops for more than six months and who did not have significant physical disabilities or severe visual impairment. However, among patients using glaucoma eye drops, some have substantial physical, cognitive, or visual impairments. Therefore, it will be necessary to expand the scope of future investigations to identify the patient populations in which this eye drop aid is most effective.

## Conclusion

The newly developed eye drop aid improved the success rate of eye drop instillation, especially in elderly individuals and those who applied eye drops in the seated position. This aid is expected to be highly useful for patients who require long-term eye drop treatment, such as those with glaucoma and for patients for whom the prevention of infection is important. Further improvements will be made to ensure appropriate eye drop treatment in the future.

## Supporting information

**S1 Video. The video demonstrates how to use the eye drop aid.**
(MP4)

## Acknowledgments

We are grateful to Kumi Kawase for her support with the current experiments. We also acknowledge Hideo Shunaga and Sayaka Masamune for their contributions to the development of the eye drop aid.

## Author contributions

**Conceptualization:** Kenji Kashiwagi.

**Data curation:** Yuka Kasai, Masako Sakamoto, Yuji Matsuda, Yuka Ito.

**Formal analysis:** Yuka Kasai, Masako Sakamoto, Kenji Kashiwagi.

**Investigation:** Yuka Kasai, Masako Sakamoto, Yuji Matsuda, Yuka Ito, Kenji Kashiwagi.

**Methodology:** Yuka Kasai, Hirotaka Haro, Kenji Kashiwagi.

**Supervision:** Hirotaka Haro, Kenji Kashiwagi.

**Writing – original draft:** Yuka Kasai, Kenji Kashiwagi.

**Writing – review & editing:** Hirotaka Haro.

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
