## [Decision Letter · Decision Letter 0]

9 Jul 2025

PONE-D-25-27512Development of a Novel Eyedrop Aid and Evaluation of Its EfficacyPLOS ONE

Dear Dr. Kashiwagi,

Thank you for submitting your manuscript to PLOS ONE. After careful consideration, we feel that it has merit but does not fully meet PLOS ONE’s publication criteria as it currently stands. Therefore, we invite you to submit a revised version of the manuscript that addresses the points raised during the review process.

The research meets the scientific and methodological standards required for publication. However, as pointed out by one of the reviewers, the manuscript would benefit from revisions to improve clarity, grammar, formatting, and overall language quality.

We ask that you carefully revise the manuscript in line with the reviewer’s suggestions, particularly in addressing awkward phrasing that may stem from automated translation and ensuring consistency in terminology and formatting throughout the text.

Please include a point-by-point response to the reviewer’s comments, outlining how each has been addressed in the revised manuscript.

We look forward to receiving your revised submission and thank you again for considering for the publication of your work.

If applicable, we recommend that you deposit your laboratory protocols in protocols.io to enhance the reproducibility of your results. Protocols.io assigns your protocol its own identifier (DOI) so that it can be cited independently in the future. For instructions see: https://journals.plos.org/plosone/s/submission-guidelines#loc-laboratory-protocols . Additionally, PLOS ONE offers an option for publishing peer-reviewed Lab Protocol articles, which describe protocols hosted on protocols.io. Read more information on sharing protocols at https://plos.org/protocols?utm_medium=editorial-email&utm_source=authorletters&utm_campaign=protocols.

We look forward to receiving your revised manuscript.

Kind regards,

Sudheesh Sreerangam Nair, B.V.Sc &A.H., M.V.Sc., Ph.D

Academic Editor

PLOS ONE

Journal Requirements: 

2. We note that you have a patent relating to material pertinent to this article. Please provide an amended statement of Competing Interests to declare this patent (with details including name and number), along with any other relevant declarations relating to employment, consultancy, patents, products in development or modified products etc. Please confirm that this does not alter your adherence to all PLOS ONE policies on sharing data and materials, as detailed online in our guide for authors http://journals.plos.org/plosone/s/competing-interests by including the following statement: ""This does not alter our adherence to  PLOS ONE policies on sharing data and materials.” If there are restrictions on sharing of data and/or materials, please state these. Please note that we cannot proceed with consideration of your article until this information has been declared.

3. We note that Figure 2 includes an image of a patient in the study.

Additional Editor Comments:

Your study presents a novel eyedrop administration aid and evaluates its effectiveness compared to existing devices. The research is original and meets the scientific and methodological standards required for publication. However, as pointed out by one of the reviewers, the manuscript would benefit from revisions to improve clarity, grammar, formatting, and overall language quality.

We ask that you carefully revise the manuscript in line with the reviewer’s suggestions, particularly in addressing awkward phrasing that may stem from automated translation and ensuring consistency in terminology and formatting throughout the text.

Please include a point-by-point response to the reviewer’s comments, outlining how each has been addressed in the revised manuscript.

We look forward to receiving your revised submission and thank you again for considering for the publication of your work.

Reviewers' comments:

Reviewer's Responses to Questions

**Comments to the Author**

1. Is the manuscript technically sound, and do the data support the conclusions?

Reviewer #1: Partly

Reviewer #2: Yes

2. Has the statistical analysis been performed appropriately and rigorously? 

Reviewer #1: I Don't Know

Reviewer #2: Yes

3. Have the authors made all data underlying the findings in their manuscript fully available?

Reviewer #1: Yes

Reviewer #2: Yes

4. Is the manuscript presented in an intelligible fashion and written in standard English?

Reviewer #1: Yes

Reviewer #2: Yes

5. Review Comments to the Author

Reviewer #1: The present report describes a new eyedrop aid and evaluates its beneficial effect on successful drops’ instillation in glaucoma patients.

The idea of using facial bony support to improve drops instillation and bottle placement is not original and some report on eyedrop aids were quoted in the present manuscript.

In the introduction the authors report on their previous publication in which “aging, a reduced cervical extension angle, a reduced ability to pick up the eyedrop container, reduced motor function of the upper limbs, motor paralysis and a reduced visual field or vision were found to be the major reasons for instillation failure”. Nevertheless, the inclusion criteria for the present report included “no upper limb loss or dysfunction, and no physical or cognitive disturbances that would interfere with daily life and visual acuity of at least 1 logMAR score in at least one eye”. The study population, therefore, does not enable demonstration of the efficiency of the proposed eyedrop aid for those who need it most (patients having manual motor dysfunction, meaningfully reduced vision and manual instability / tremor). As all eyedrop aids holds the tip of the bottle far enough from the ocular surface in order to prevent/reduce corneal or periocular touch, some of them reduce the squeeze power needed for drop instillation and most of them fit to more than one drops bottle the only theoretical advantage of the current aid is that it can assist in cases of impaired neck retroflexion.

In the methods section instillation failure was defined as “the tip of the eyedrop bottle directly touched the surface of the eye, eyelid, eyelashes or face or more than two eyedrops were applied in a single trial”. One can assume that the definition of failure in the methods section allows for the use of up to two drops in order to achieve successful instillation for cases in which the first drop did not land in the conjunctival sack.

In the results section the authors wrote:” Among the 25 patients in the instillation failure group, 12 (48%) experienced eyedrops landing around the eye, whereas in 6 patients (24%), the tip of the eyedrop bottle made contact with the eyelashes. No patients administered more than two drops at a time”.

Cases in which a drop landed outside the conjunctival sack are those who needed second drop but are not defined as failure. In addition, no reason is given for failure in another 7 participants.it seems that the actual failure rate and its associations need to be re-calculated.

The presented aid looks convenient for use, can be easily cleaned and re-used, is reported to be not expensive (but not very cheap either) and is especially efficient for patients with reduced cervical extension angle. In the current report it was not tested for patients who need it most and the comparative assessment of its efficiency is not well established based on the definition given by the authors.

Although the present aid looks promising the fact it was not tested on those who really need it along with the unclear criteria for instillation failure definition and the consequent inability for actual assessment of the advantage of the use of this aid calls for additional data as well as better data analysis before it can be published.

Reviewer #2: The authors have developed a new eyedrop administration aid and are examining its usefulness in comparison with previously reported reports. Although the content itself has met the criteria for publication, there are several awkward parts that might have been translated by an automatic translator, so it would be desirable to revise them.

Introduction

You may want to add a connective word like 'However,' to the sentence starting with 'Many patients'.

You may want to add a connective word like 'In an effort to resolve this issue' to the sentence starting with Many eyedrop instillation aids.

You have to add a colon (:) after [12-16].

Patients

The sentence starting with 'With respected to corrected visual acuity' should be changed to 'With respect to'.

Discussion

You have to omit " at the third paragraph.

You may want to change 'However' to 'In addition' in the last sentence of the fourth paragraph.

You have to change 'only an eye drop aid' to 'only an eyedrop aid' to align with the terminology employed in other parts of the manuscript.

You have to omit " at the end of the Discussion.

6. PLOS authors have the option to publish the peer review history of their article (what does this mean? ). If published, this will include your full peer review and any attached files.

**Do you want your identity to be public for this peer review?** For information about this choice, including consent withdrawal, please see our Privacy Policy .

Reviewer #1: No

Reviewer #2: **Yes: ** YUICHI ASAHINA

---

## [Author Response · Author response to Decision Letter 1]

19 Jul 2025

Dear Reviewers,

We sincerely thank you for your thoughtful reviews.

Please refer to the attached file for our responses to your comments.

---

## [Decision Letter · Decision Letter 1]

1 Sep 2025

Development of a Novel Eyedrop Aid and Evaluation of Its Efficacy

PONE-D-25-27512R1

Dear Dr. Kashiwagi,

We’re pleased to inform you that your manuscript has been judged scientifically suitable for publication and will be formally accepted for publication once it meets all outstanding technical requirements.

Kind regards,

Sudheesh Sreerangam Nair, B.V.Sc &A.H., M.V.Sc., Ph.D

Academic Editor

PLOS ONE

Additional Editor Comments (optional):

Reviewer #3:

Reviewer #4:

Reviewers' comments:

Reviewer's Responses to Questions

**Comments to the Author**

1. If the authors have adequately addressed your comments raised in a previous round of review and you feel that this manuscript is now acceptable for publication, you may indicate that here to bypass the “Comments to the Author” section, enter your conflict of interest statement in the “Confidential to Editor” section, and submit your "Accept" recommendation.

Reviewer #3: (No Response)

Reviewer #4: All comments have been addressed

2. Is the manuscript technically sound, and do the data support the conclusions?

Reviewer #3: Partly

Reviewer #4: Yes

3. Has the statistical analysis been performed appropriately and rigorously? 

Reviewer #3: Yes

Reviewer #4: Yes

4. Have the authors made all data underlying the findings in their manuscript fully available?

Reviewer #3: Yes

Reviewer #4: Yes

5. Is the manuscript presented in an intelligible fashion and written in standard English?

Reviewer #3: Yes

Reviewer #4: Yes

6. Review Comments to the Author

Reviewer #3: Methods

Patients

“inclusion criteria: no upper limb loss or dysfunction “ Were patients suffering any tremors included n the study? Or were “tremors” considered “upper limb dysfunction” and these patients excluded from the study?

“the patients had a logMAR score of at least 1 in at least one eye” 1. Was a reasonable visual acuity in the “experiment eye” an inclusion criterion? In other words, was an eye included in the study that had e.g. a no PL vision? 2. Was the residual vision location considered in study inclusion? In other words, patients with a residual tubular field in a temporal island of vision, in which straight ahead gaze was scotomatous, were these included in the study? Was their performance different from patients with intact straight ahead gaze?

Was the experimental design made for both eyes of the patient or only one eye? Which eye? If both eyes, was the order of instillation randomized? Was the handedness of the patients taken into account or documented with the results (patients may find difficulty instilling eyedrops in the eye on the side of the non-dominant hand or may not be able to switch hands to instill the eyedrops in the other eye)?

Results

“Five patients experienced instillation failure due to the administration of eye drops to the periocular area” Did these patients fail to apply the Aid to the correct position? Or was “periocular” spilling associated with he Aid in the proper position? Please make this clear.

“whereas in 6 patients (24%), the tip of the eye drop bottle made contact with the eyelashes.” Did these 6 patients demonstrate eyelashes touch with or without the Aid? If with the aid, please explain how this happened when the Aid mechanically prevents this contact.

Conclusion

“This aid is expected to be highly useful for patients who require long-term eye drop treatment,” Please refrain from using abstract terms such as “highly” useful, “useful” is more in line with the study findings.

References

Outdated, with only 15% of the references in the past 5 years. Please cite more recent references

Reviewer #4: The revised manuscript content reads well, and will be of interest to the reader.

Formatting may need to be improved in the final submitted document.

7. PLOS authors have the option to publish the peer review history of their article (what does this mean? ). If published, this will include your full peer review and any attached files.

**Do you want your identity to be public for this peer review?** For information about this choice, including consent withdrawal, please see our Privacy Policy .

Reviewer #3: No

Reviewer #4: No

---

## [Editor Report · Acceptance letter]

PONE-D-25-27512R1

PLOS ONE

Dear Dr. Kashiwagi,

I'm pleased to inform you that your manuscript has been deemed suitable for publication in PLOS ONE. Congratulations! Your manuscript is now being handed over to our production team.

Kind regards,

on behalf of

Dr. Sudheesh Sreerangam Nair

Academic Editor

PLOS ONE